# Combined Treatments Consisting of Calcium Hydroxide and Activate Carbon for Purification of Xylo-Oligosaccharides of Pre-Hydrolysis Liquor

**DOI:** 10.3390/polym11101558

**Published:** 2019-09-25

**Authors:** Feng Xu, Jiachuan Chen, Guihua Yang, Xingxiang Ji, Qiang Wang, Shanshan Liu, Yonghao Ni

**Affiliations:** 1State Key Laboratory of Biobased Material and Green Papermaking, Qilu University of Technology, Shandong Academy of Sciences, Jinan 250353, China; xufeng19191@gmail.com (F.X.); chenjc@qlu.edu.cn (J.C.); jxx@qlu.edu.cn (X.J.); wangqiang83@qlu.edu.cn(Q.W.); liushanshan@qlu.edu.cn(S.L.); 2Limerick Pulp and Paper Centre, University of New Brunswick, Fredericton, New Brunswick E3B 5A3, Canada; yonghao@unb.ca

**Keywords:** xylo-oligosaccharides, pre-hydrolysis liquor, calcium hydroxide treatment, activated carbon treatment, purification

## Abstract

In this study, the effect of a combined treatment consisting of calcium hydroxide (CH) followed by activated carbon (AC) on the purification of hemicellulose in the pre-hydrolysis liquor (PHL) from pulping process has been evaluated. The results show that lignin and furfural of PHL are efficiently removed, and the lignin removal is achieved by forming complexes onto CH particles in the CH treatment process, while acetic acid (acetate) is formed from the hydrolysis of acetyl groups present in the dissolved hemicelluloses in the PHL. The loss of xylo-oligosaccharides (XOS) is moderate, even at a high CH dosage of 0.8% while the xylose concentration is essentially unchanged. For the AC treatment, the optimal treating pH can enhance the interactions between AC and residual lignin and change the zeta potential of AC resulting in improved lignin adsorption onto AC. An increase of AC dosage has the tendency to adsorb more XOS_DP>6_ than XOS_DP2~6_. Overall, 66.9% of lignin and 70.1% of furfural removals are achieved under the optimal conditions of CH and AC treatment process, with a 5.9% total xylosugars loss. The present combination of CH and AC treatment process was more effective and selective for purification of xylosugars of PHL.

## 1. Introduction

As a significantly global strategy, the biorefinery concept gets a perfect embodiment and application in platform chemicals, biological materials, and biological energy. In this concept, it is imperative to make a sustainable and environmentally friendly utilization for biomass resources, especially lignocellulosic materials. Recently, more attention has been focused on the integration of forest biorefinery [1,2]. As a vital aspect for the implementation of integrated forest biorefinery, separation of hemicellulose from wood chips by hot water/steam pre-extraction/hydrolysis prior to the process of kraft-based dissolving pulping can offer a potential opportunity to obtain valued-added lignocellulosic products.

It is well known that hemicelluloses saccharides have wide applications in many fields. They can be transformed into fuel ethanol, levulinic acid, xylitol, etc. [3,4,5]. Especially, xylo-oligosaccharides (XOS) are also considered as important platform chemicals to produce value-added products due to their diverse physiological functions, such as lowering cholesterol, improving bowel function, and reducing the risk of colon cancer [6,7,8]. However, contaminants present in PHL, such as lignin, furfural, and hydroxymethylfurfural (HMF), hinder the separation and utilization of hemicelluloses saccharides [9,10,11]. For example, lignin can abate the purity and yield of hemicellulose products and reduce the added-value of downstream products. Thus, it is necessary to find an effective pathway to remove the contaminants from PHL and purify xylose and XOS in PHL. 

Various methods have been applied for separating/isolating lignin from PHL. For example, the influence of acidification/polyethylene oxide (PEO) flocculation on the lignin removal was investigated by Shi et al. [12]. They reported that 22% of the lignin was removed from PHL at pH = 2 and PEO level of 350 mg/L with non-obvious loss of sugar. However, Yasarla and Ramarao thought that the flocculation was not specific to the removal of lignin because polydiallyldimethylammonium chloride (p-DADMAC) removed 70.34% of lignin with significant 36.77% sugar loss [13]. In contrast to flocculation treatment, membrane filtration can effectively remove the lignin, but their problems such as high cost, pollution, and difficult regeneration hinder severely their industrial application [14]. Moreover, acidification was reported as an economical method for removing lignin from PHL, but it depends on the properties of the lignin in the PHL, such as molecular weight, structures, and compositions [15]. Previous study showed that approximately 50.0% of lignin and 17.1% of hemicellulose were removed from PHL via adding H_2_SO_4_ to pH = 2 [16]. Thus, it is essential to improve the selectivity between lignin and hemicellulose. 

Lime treatment is an effective way to remove contaminants from PHL [17]. For example, Wang et al. reported that lime treatment and neutralization removed 44.2% of contaminants with only 5.6% of hemicellulose loss at 0.5% lime level [18], and the selectivity of lignin was higher at lower lime dosage. But the hemicellulose compositions may tend to be adversely degraded into organic acids under hot alkaline conditions at a higher lime dosage. Therefore, in this study, calcium hydroxide (CH) was selected to avoid heat-degradation of hemicellulose because CH releases negligible energy when dissolved in PHL. As the first objective of this work, the effect of CH treatment on the contents of contaminants (i.e., lignin and furfural) and xylosugars, especially XOS, with different degree of polymerization (DP) were investigated. In addition, the mechanism of lignin removal in detail was identified. 

Some lignin still exists in CH-treated PHL after CH treatment. Activated carbon (AC) adsorption was selected to remove the residual lignin due to several advantages such as remarkable adsorption capacity, simplicity in operation, and environmental friendliness [19,20]. Effects of surface characteristics of AC such as its surface area, pore size distribution, and chemical groups on the adsorption capacity were reported [19,21]. Different adsorbates (i.e., lignin, acetic acid, furfural, monosaccharide, and oligosaccharide) in the PHL or in the model PHL and their adsorption performance on AC were investigated by Shen et al. [22] and Fatehi et al. [23]. For the treatment of aromatic compounds, Bautistatoledo et al. investigated the influence of AC adsorption on bisphenol A (2-2-bis-4-hydroxypheniyl propane) from water and concluded that the adsorption capacity of AC to bisphenol A depends on solution pH [21]. It is well known that diverse lignin degradation byproducts in PHL including oligomer, dimer, and monomer are all aromatic compounds rich in phenolic hydroxyl [24]. Therefore, effect of treating pH on the capacity of AC adsorbing lignin and total xylosugars in PHL was explored experimented in this work. 

It is reported that the high loss of hemicellulose was mainly caused by XOS loss [2,25,26]. In order to maximize the removal of contaminants and minimize the loss of XOS, the adsorption efficiency of AC and relevancy analysis between the DP of XOS and the loss of XOS in AC treatment process at different AC dosages are necessary to investigate. Thus, the adsorption behavior of AC to contaminants, xylose, and XOS with different DP was experimented in this work.

The present study introduces a purification method consisting of CH treatment, followed by AC adsorption to purify the pre-hydrolysis liquor (PHL) so that xylose and XOS in PHL can be further utilized for the production of value-added products. Effects of CH treatment on the removal of contaminants and loss of xylosugars were investigated. The influence of solution pH and AC dosage on adsorption of contaminants and XOS with different DP was especially evaluated.

## 2. Materials and Methods 

### 2.1. Materials

The fast-growing poplar wood chips were taken from Sun Paper Co. LTD in (Yanzhou, China) Shandong province, China. Commercial activated carbon (AC, wood-based, powder) with 200 mesh particle size was supplied by Guangzhou Haiyan Company (Guangzhou, China). Calcium hydroxide (Ca(OH)_2_, powder, 99%), sodium hydroxide (NaOH, 99%), sulfuric acid (H_2_SO_4_, 99%), and phosphoric acid (H_3_PO_4_, 85%) were purchased from Tianjin Hengxing Company, Tianjin, China.

### 2.2. PHL Preparation

The poplar wood chips were washed with water to remove dirt and air-dried prior to the hot-water pre-hydrolysis process. Pre-hydrolysis was carried out in a digester (KRK, No.2611, Kumagai Riki Kogyo Ltd., Tokyo, Japan). The hot-water pre-hydrolysis conditions were 1:6 of ratio of solid to liquid and 170 °C for 60 min. PHL was obtained from the digester for analysis, and the solid residual chips were used to produce dissolving pulp.

### 2.3. Calcium Hydroxide (CH) Treatment

Ca(OH)_2_ powder existed in two forms: (1) undissolved CH particles and (2) dissolved CH (Ca^2+^ in solution) after added into PHL. Effect of CH forms on lignin removal during CH treatment process was investigated in the same pH. CH treatment means treatment using undissolved CH particles, and CH solution treatment means treatment using dissolved CH. Different CH form can remove lignin in different approach, such as the adsorption of lignin onto CH particle surface or forming precipitated complexation between lignin and calcium ions. The conditions of CH treatment were room temperature and the dosages of CH powder/solid varying from 0.2% to 2.0% (based on the weight of the original PHL). A total of 50 g of the original PHL was mixed with the CH powder for 10 min. Then the mixture was centrifuged at 5000 rpm for 5 min. The precipitation (containing CH particles) was denoted as adsorbed lignin and collected for Fourier transform infrared spectroscopy (FTIR) analysis while supernatant liquid was collected as CH-treated PHL and stored at 4 °C for analysis.

To explore the effect of dissolved CH solution on lignin removal in CH treatment process, different qualities CH solution (pH = 13), obtained via dissolving CH powder in deionized water, were mixed with original PHL for 10 min to obtain different pH values. Then the mixture was centrifuged at 5000 rpm for 5 min. The precipitates were denoted as complexed lignin and collected for FTIR analysis while the supernatant liquid was collected for lignin removal analysis. Additionally, to study the changes of functional groups of the lignin before and after CH treatment, the original PHL was centrifuged at 5000 rpm for 5 min, and the precipitates were denoted as PHL lignin and collected for FT-IR analysis. 

### 2.4. Activated Carbon (AC) Treatment

The CH-treated PHL was mixed with AC at AC dosage of 0.4% (based on the weight of CH-treated PHL and our previous optimal dosage) under varied treatment solution pH (10.2, 7, 6, 5, 4, and 3.6) to optimize the pH for improving AC adsorption efficiency. In addition, effects of varying AC dosage on lignin removal and total xylosugars loss of the CH-treated PHL under the optimal pH were optimized. All experiments were carried out at room temperature and the mixture was shaken at 150 rpm for 10 min. Then the mixtures were centrifuged at 5000 rpm for 5 min and the filtrates were collected as AC-treated PHL. Phosphoric acid was used to adjust the pH from 10.2 to 7 to eliminate calcium ions influence. 

### 2.5. Chemical Composition Analyses

The concentrations of xylose and XOS_DP2~6_ with 2~6 DP (including xylobiose, xylotriose, xylotetraose, xylopentaose, and xylohexaose) of the PHL samples unhydrolyzed by acid were directly measured by an ion chromatography (ICS-5000+, Thermo Scientific, Inc., Sunnyvale, CA, USA), which was equipped with a Dionex CarboPac PA200 column (3 mm × 250 mm, Thermo Scientific Inc., Sunnyvale, CA, USA) and an ED 40 electrochemical detector. The mobile phases were 0.1 M NaOH and 0.5 M NaOAc containing 0.1 M NaOH at a flow rate of 0.3 mL/min and column oven temperature of 30 °C [27]. For determining total xylosugars concentration of PHL, the collected PHL sample was carried out by acid hydrolysis under 4 wt % of H_2_SO_4_ dosage at 121 °C for 60 min [28] and followed by ion chromatography analysis to obtain the concentration of total xylosugars. XOS included XOS_DP2~6_ and XOS_DP>6_ with more than 6 DP. The concentration of XOS in PHL was determined by mass balance (Equation 1):(1)XOS (g/L)=Total xylosugars−Xylose
where total xylosugars is the concentration of total xylosugars from the acid hydrolyzed PHL (g/L), xylose is the concentration of xylose from the PHL samples unhydrolyzed by acid (g/L).

The concentration of XOS_DP>6_ in PHL was expressed in Equation 2,
(2)$${\mathrm{XOS}}_{\mathrm{DP}>6}\;\left(g/L\right)=\mathrm{XOS}-\frac{{\mathrm{XOS}}_{\mathrm{DP}2\sim 6}}{0.88}$$
where XOS is the concentration of XOS from Equation (1), XOS_DP2~6_ is the concentration of XOS_DP2~6_ from the PHL samples unhydrolyzed by acid, and 0.88 was conversion of constant when XOS_DP2~6_ were converted to xylose [28]. 

The concentrations of acetic acid and furfural were measured using a Shimadzu LC-20T high performance liquid chromatography (HPLC) system (Kyoto, Japan), which was equipped with a SUPELCOGELC-610H column (30 cm × 7.8 mm, Sigma-Aldrich, St. Louis, MO, USA) and the SPD-20A detector. Samples were run at 30 °C and eluted at 0.7 mL/min, and mobile phase was 0.1% H_3_PO_4_ (*v*/*v*). 

The concentration of lignin in solutions was determined at wavelengths of 205 nm using a UV/vis spectrophotometer (Agilent Technologies, Palo Alto, CA, USA) according to the TAPPI standard (TAPPI UM 250) [29].

The data of all analysis was the average of duplicates with an average relative standard deviation within 5%. 

### 2.6. Zeta Potential of AC

Zeta potential of AC was measured using a dynamic light scattering (DLS) analyzer (Zetasizer Nano ZS90, Malvern Instruments Ltd., London, UK). A total of 0.1g of AC was added to 20 mL of 50 mM NaCl solution at different pH. Then the mixtures were shaken at 25 °C for 24 h in a shaker. A volume of 1 mL of mixtures was diluted decuple with 50 mM NaCl solution to ensure a good conductivity of the solution. The zeta potential of each sample was the average of triple measurements.

### 2.7. AC Characterization

The acid and basic groups on the AC surface were determined by following a literature method [30]: This method was based on acid/base titration of carbon acidic or basic centers. First, 0.1 g of AC was added to 20 mL of 50 mmol/L NaHCO_3_, Na_2_CO_3_, NaOH, or HCl solutions, and the mixtures were shaken for 24 h at room temperature. Then, 5 mL of supernatant was titrated with 50 mmol/L of HCl or NaOH solution at room temperature. It showed that functional groups of AC rich in acidic groups mainly included phenolic groups, lactonic groups, and carboxylic groups whose content were 0.61, 0.35, and 0.04 mmol/g, respectively. 

### 2.8. FTIR Analyses

The surface functional groups of lignin were characterized using a Fourier transform infrared spectrometer (VERTEX70, Bruker, Karlsruhe, Germany). The oven-dried samples were embedded in KBr pellets in a mixture of about 1 wt % KBr. The spectra were monitored in a transmittance band mode in the range 400–4500 cm^−1^.

## 3. Results and Discussion

### 3.1. PHL Chemical Compositions

In the pre-hydrolysis process, xylan is degraded into XOS and xylose [2]. The purpose of CH and AC treatment is to separate efficiently the impurities (lignin and furfural) in the PHL for the further utilization of XOS with different DP. For this work, XOS is defined as the sum of XOS_DP2~6_ and XOS_DP>6_. XOS_DP2~6_ includes xylobiose, xylotriose, xylotetraose, xylopentaose, and xylohexaose. XOS_DP>6_ represents XOS with DP higher than 6. The chemical compositions of the PHL are listed in Table 1. It shows that XOS and lignin were mainly compositions in PHL from hardwood dissolving pulp production processes.

### 3.2. CH Treatment

#### 3.2.1. Effect of Different CH forms

The mechanism of lignin removal during CH treatment process was demonstrated in Figure 1. Lignin in the original PHL remained relatively stable due to abundant phenolic hydroxyl (PhOH) [24], while the addition of CH affected the stability of lignin in the PHL, which resulted in lignin destabilization for the formation of CH precipitates [31]. It is well known that the CH exists in two forms in the PHL [17,18]: (1) undissolved CH particles and (2) dissolved CH (Ca^2+^ in solution). As a result, the adsorption of lignin onto the CH particle surface (Figure 1A) and the precipitated complexes between lignin and calcium ions (Figure 1B) were proposed.

The significant lignin removal was obtained in CH treatment in previous research work [2]. Solubility of Ca(OH)_2_ dissolved in PHL varied from 32.5% to 9.8% when Ca(OH)_2_ dosage ranged from 0.2% to 2.0%, and most of Ca(OH)_2_ added in PHL existed in solid form. Effect of CH forms and pH on lignin removal in CH treatment process are listed in Table 2. It can be seen from Table 2 that, for the CH treatment process, the lignin removal significantly increased from 6.7% to 37.1% with the increase of pH from 6.6 to 11.0 due to the lignin absorption onto calcium hydroxide particles, and the formation of lignin-calcium complexes. While for the CH solution treatment process, the lignin removal increased slowly from 3.1% to 5.0% with the increase of pH from 6.6 to 9.0. The lignin removal is due to the formation of lignin-calcium complexes resulting from the electrostatic association of ionized phenolic lignin and calcium ions [18,31]. When increasing pH from 9.0 to 10.0, the lignin removal obviously increased from 5.0% to 9.2%. This is attributed to the formation of lignin-calcium complexes (Figure 1B). Subsequently, the lignin removal had no obvious change by further increasing the pH to 12. Thus, alkaline condition was suitable for CH solution treatment because the phenolic lignin tends to destabilize and precipitate [31]. At the same treatment pH of 11.0, the lignin removal was 9.8% in the CH solution treatment process, while the lignin removal was 37.1% in the CH treatment process. Evidently, the CH treatment had higher efficiency than that of CH solution treatment on the lignin removal. The adsorption of lignin onto undissolved CH particles may be the dominant mechanism for lignin removal in the CH treatment process (Figure 1A).

#### 3.2.2. FTIR Analyses of Separated Lignins

The lignin from the CH treatment process, CH solution treatment process and original PHL were analyzed by FTIR (Figure 2). The adsorbed lignin means the lignin adsorbed on undissolved CH particles after the CH treatment process, the complexed lignin means the lignin precipitates obtained by CH solution treatment, and the PHL lignin is obtained by centrifugation of the original PHL. The adsorbed lignin, complexed lignin, and PHL lignin showed classic structures [16,32], for example, 1520 cm^−1^ is associated to the vibration of C=C groups of the aromatic skeletal of lignin. 1425 cm^−1^ attributed to C–H in plane deformation with aromatic ring stretching, 1330 cm^−1^ attributed to C–O of the syringyl ring. The FTIR spectra of original CH particles were similar to those reported in the literature [2]. Those FTIR spectra at 1520, 1425, and 1330 cm^−1^ for the CH precipitation adsorbed by lignin, lignin-calcium complexes, and original PHL lignin proved the adsorption of lignin onto the undissolved CH particles.

#### 3.2.3. Effect of CH Dosage

The effect of CH dosage on PHL compositions is shown in Figure 3. It can be seen from Figure 3A that the removal of contaminants was strongly affected by the CH dosage. The contaminants removal increased rapidly when the CH dosage increased from 0 to 0.6%, and the removal of lignin and furfural was 34.3% and 55.5% respectively at 0.6% CH dosage. Lignin was removed via the formation of lignin-calcium complexes and the absorption of lignin onto undissolved CH particles [17,18]. Further increasing the CH dosage to 2%, the lignin removal changed slightly, but furfural was almost completely removed, which was due to furfural can be further oxidized under alkaline conditions [17]. Oppositely, the acetic acid concentration increased from 1.55 to 3.50 g/L as the CH dosage increased from 0 to 0.6%, the reason being the cleavage of acetyl groups of XOS in PHL under the alkaline conditions [20,22]. Acetic acid produced by deacetylation could be extracted and recovered [33,34]; what is more, deacetylation can provide more suitable conditions for the production of XOS especially XOS_DP2~6_ by enzymatic hydrolysis [35]. Deacetylation can make the xylanase more accessible to xylan during enzyme hydrolysis and further enhance the efficiency of enzymatic hydrolysis.

Figure 3B showed that the concentration of total xylosugars was unchanged when the CH dosage was lower than 0.6%. Further increasing the CH dosage to 2%, the loss of xylose, XOS (including XOS_DP2~6_ and XOS_DP>6_) and total xylosugars reached up to 8.2%, 42.5%, and 50.7% respectively, the reason might be alkaline-degradation and adsorption [18]. As shown, the loss in XOS was much more in CH treatment process than the xylose loss due to XOS with higher molecular weight than xylose and easily adsorbed onto CH particles, which was in accordance with previous studies [2,17]. 

In addition, Figure 3B showed that the loss of XOS_DP2~6_ was slightly higher than that of XOS_DP>6_ when increasing the CH dosage from 0.6% to 0.8%, this was due to the higher concentration of XOS_DP2~6_ than XOS_DP>6_ in the original PHL (Table 1). Whereas, further increasing the CH dosage from 1% to 2%, the loss of XOS_DP>6_ increased from 8.9% to 22.2% while XOS_DP2~6_ increased from 8.7% to 18.3%. The higher loss of XOS_DP>6_ than XOS_DP2~6_ was probably ascribed to the degradation of XOS, especially XOS_DP>6_ could generate XOS_DP2~6_ by the peeling reaction under mild alkaline conditions. Furthermore, a part of XOS_DP>6_ was probably removed due to their adsorption onto undissolved CH particles. This was in accordance with the notion that xylan with higher molecular weight has a lower solubility in aqueous solution than xylose, thus easier to be adsorbed onto AC [23]. CH particles have specific surface area like the AC. There is a contrary charge density between XOS_DP>6_ and CH particles; consequently, an electrostatic charge interaction formed, and the adsorption of XOS_DP>6_ on CH particles was higher than XOS_DP2~6_ and xylose. 

The optimal dosage of CH was 0.6%, based on the minimum loss in total xylosugars (about 1%) and the maximum contaminants removal (34.3%).

### 3.3. AC Treatment

#### 3.3.1. pH Value

AC was selected to remove the residual lignin with lower molecular weight after the CH treatment. The pH can influence AC absorption efficiency. The effect of pH on lignin removal and total xylosugars loss in the CH-treated PHL at 0.6% of CH dosage are shown in Figure 4. It can be seen from Figure 4 that the lignin removal at 10.2 of pH was 35.1%, and then increased to 39.6% at 7 of pH. Subsequently, with further decreasing pH to 3.6, the lignin removal reached up to 60.7%. The reason for lignin removal might be attributed to π–π dispersion interactions, which were mainly influenced by electrostatic interaction between lignin and AC [30,36].

For the residual phenolic lignin in the CH-treated PHL, its ionizing degree decreased with the decrease of pH. Additionally, Figure 4 showed that the net negative charge of AC surface obviously decreased when the pH decreased from 10.2 to 3.6. As a result, the electrostatic repulsion between the AC and the lignin decreased due to the decrease of negative charges of acidic groups on the AC surface, which can enhance the interactions between AC and residual lignin, consequently, resulting in more lignin removal.

Similar to lignin removal, the loss of total xylosugars was lower when pH was higher than 5. Then with further decreasing the pH to 3.6, the loss of total xylosugars reached up to 9.1% (Figure 4), which related to the electrostatic interactions between AC and XOS. The ionizing of acidic groups such as uronic acids in XOS increased the negative charge of XOS under alkaline conditions, which led to the stronger electrostatic repulsion between XOS and AC, and thus the total xylosugars loss decreased. At an acidic pH, more XOS loss occurred due to the decreased electrostatic repulsion.

The optimal treatment pH in AC treatment process was 5, which leads to further 49.5% of the lignin removal and 5.2% of total xylosugars loss in the CH-treated PHL.

#### 3.3.2. AC Dosage 

AC dosage is an important factor in AC treatment process. The CH-treated PHL at 0.6% of CH dosage still contain 3.33 g/L of lignin and 0.28 g/L of furfural and need to be further treated with AC for removing the residual contaminants from PHL, and the AC dosages varied in 0.2%, 0.4%, 0.6%, 0.8%, 1.0%, and 1.2% at the pH of 5. The influences of AC dosage on the compositions of the CH-treated PHL are presented in Figure 5. Figure 5 showed that the lignin removal increased obviously when AC dosage was below 0.4%, and then a slight increase appeared with AC dosage more than 0.4%. This means that the selectivity of AC adsorption is better at lower AC dosages. However, a higher AC dosage, such as 1.2%, was not advisable mainly due to the 22.5% of total xylosugars loss (1.5% of xylose loss and 21.0% of XOS loss), although 73.1% lignin removal and 72.0% furfural removal were obtained. The competitive adsorption between lignin and XOS may result in a higher loss of XOS when the concentration of XOS was obviously higher than lignin and furfural in the CH-treated PHL. In addition, the removal of lignin was consistently higher than that of furfural when the AC dosages were below 1.2%. Fatehi et al. reported that the adsorption behavior of the materials was affected by the molecular structure and weight [23]. Thus, lignin and furfural in the CH-treated PHL, as hydrophobic materials, were easily adsorbed into the micropores of AC due to its lower molecular weight compared to CH successfully removing lignin with larger molecular weight in CH treatment. Besides, the content of the residual lignin of 3.33 g/L was evidently higher than the residual furfural of 0.28 g/L after CH treatment, therefore, the diffusion of residual lignin into the pores of AC was probably efficient compared to the residual furfural, which facilitated the adsorption of lignin.

Figure 5 shows that the loss of xylose, XOS_DP2~6_, XOS_DP>6_, and total xylosugars increased gradually, and the loss of XOS_DP>6_ was higher than XOS_DP2~6_ and xylose with the increase of AC dosage. Compared to the hydrophobic lignin and furfural, xylosugars (XOS_DP>6_, XOS_DP2~6_, and xylose) in PHL were more hydrophilic; XOS_DP>6_ have the lower solubility than XOS_DP2~6_ and xylose due to its higher molecular weight, and are easily adsorbed by AC, which resulting the higher loss of XOS_DP>6_ (Figure 5). The conclusion is consistent with the literature in which xylan was more easily adsorbed onto AC than xylose due to its lower solubility and higher molecular weight as well as a tendency for separation from solutions [23]. Therefore, a decrease of the polymerization degree of XOS was necessary for the maximum lignin removal and the minimum XOS loss. AC treatment can be considered as one of efficient approaches for xylose and XOS purification and separation in industrial biorefining practice. For the CH-treated PHL, the optimal conditions of AC treatment were 0.4% of AC dosage and 5 of treatment pH. 

### 3.4. Combined CH and AC Treatment

Table 3 shows the effect of combined AC and CH treatment on the removal of lignin and furfural and the loss of total xylosugars. As seen as in Table 3, in the second CH treatment stage of the AC + CH treatment sequence, lignin removal increased from 48.3% to 49.2% when the dosage of CH increased from 0.4% to 0.6%. Compared to CH + AC treatment sequence, the efficiency of CH treatment on lignin removal decreased in the AC + CH treatment sequence, indicating that the CH treatment is more suitable to be placed ahead of the AC treatment in the contaminant removal process because CH cannot efficiently remove the residual lignin with lower molecular weight in AC-treated PHL. While in the CH + AC treatment sequence, 66.9% of lignin removal can be obtained because the CH treatment could efficiently remove lignin from original PHL due to the formation of adsorption and lignin complex. The combination treatment had only slight effect on the total xylosugars loss.

### 3.5. PHL Chemical Compositions in Different Treatment Stages

The concentrations of xylose, XOS, total xylosugars, lignin, acetic acid, and furfural of the PHL at different treatment stages are shown in Table 4. The CH treatment removes 34.3% lignin and 55.5% furfural at 0.6% of CH dosage, while the loss of total xylosugars is negligible, and the concentration of acetic acid increases by about 2.3 folds. AC treatment led to additional 32.6% of the lignin removal and 5.2% of total xylosugars loss at 0.4% of AC dosage, with negligible effect on acetic acid removal. For the treatment of CH combined with AC, the removals of lignin and furfural based on original PHL were 66.9% and 70.1%, respectively, while the losses of xylose, XOS, and total xylosugars were 4.0%, 6.2%, and 5.9% respectively. The results in the literature showed that the application of polydimethyl diallyl ammonium chloride (PDADMAC) caused 70.3% lignin removal and 36.8% sugar removal [13]. In another study, 50.0% lignin and 17.1% hemicellulose were removed from PHL via the acidification process (adding H_2_SO_4_ to 2 of pH) [16]. Comparatively, the present combination of CH and AC treatment process was more effective and selective. 

### 3.6. A Proposed Process for XOS Purification

A proposed process diagram for purifying or separating xylose and XOS of PHL from the kraft-based dissolving pulp production process is shown in Figure 6. In this proposed process, CH treatment can effectively remove furfural and lignin with higher molecular weight at a negligible xylosugars loss, and AC treatment can successfully remove the residual lignin with lower molecular weight and result in a slightly loss of xylosugars. In CH treating process, CH particles remove lignin via the precipitation (lignin-calcium complexes) and adsorption; as a byproduct, the removed lignin can be further recycled and purified through acidification and crystallization for purification and improvement [31]. The used CH particles can be recycled and transformed into CaO via calcination in the lime kiln, and CaO can be further reused for the CH treatment process. The pH can be optimized for the CH-treated PHL before AC adsorption due to its significant influence on lignin adsorption efficiency onto AC surfaces. For the AC treatment, the contaminants can be further removed via adding AC at different dosage into CH-treated PHL at the optimal pH with small total xylosugars loss, and the used-AC can be regenerated via burn/thermalization and reactivated via acidification to promote the reuse and recycle of AC in the AC adsorption process [22]. Besides, the residual lignin with lower molecular weight in PHL after CH and AC treatment can be further removed via other methods, such as the laccase-induced lignin polymerization process, lignin peroxidase-induced polymerization process, and flocculation. The concentrations of xylose and XOS in PHL after purification can be increased by subsequent acid/enzymatic hydrolysis [37,38]. For the proposed process, the removal of contaminants (lignin, furfural) and the purification of hemicellulose in PHL can enhance the hemicellulose hydrolyzates obtained from the proposed process to prepare the value-added products (e.g., XOS_DP2~6_, bioethanol, xylitol, additives, dietary supplements, and chemicals) in the subsequent biorefinery process [2,7,17,39,40,41,42].

## 4. Conclusions

A combined treatment consisting of calcium hydroxide (CH) and active carbon (AC) can efficiently separate lignin and furfural of pre-hydrolysis liquor (PHL) from pulping process for the preparation of PHL containing rich xylose/ oligosaccharides (XOS), which can be further processed to produce value-added products. CH treatment removes lignin by forming lignin-calcium complexes and lignin absorption onto CH particles, while the acetic acid concentration increases due to the hydrolysis of acetyl groups. The pH during the AC treatment has significant influence on the adsorption efficiency of lignin onto AC; a high AC dosage will adsorb more XOS_DP>6_ than XOS_DP2~6_. The optimal conditions of AC treatment are 0.4% of AC dosage and 5 of pH. Under the optimal conditions of combined treatment process, 66.9% of lignin and 70.1% of furfural removals are achieved, with a 5.9% of total xylosugars loss. Due to the efficient removal of contaminants, low total xylosugars loss, and good selectivity, the combined treatment consisting of CH followed by AC is suited for the purification of hemicellulose in PHL to enhance the hemicellulose hydrolyzates to prepare the value-added products in the subsequent biomass biorefinery process. 

## Figures and Tables

**Figure 1 polymers-11-01558-f001:**
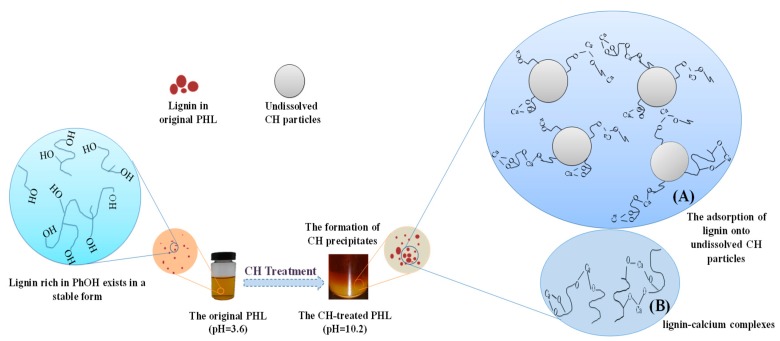
Schematic diagram of the formation of CH precipitates in two forms of lignin-calcium complexes and the adsorption of lignin onto CH particles during CH treatment process. (**A**) means that the adsorption of lignin onto undissolved CH particles mainly occurs during CH treatment process due to most of Ca(OH)_2_ existing in solid form in PHL. (**B**) means lignin-calcium complexes between lignin and calcium ions were limited to form based on the lower solubility of Ca(OH)_2_ in PHL.

**Figure 2 polymers-11-01558-f002:**
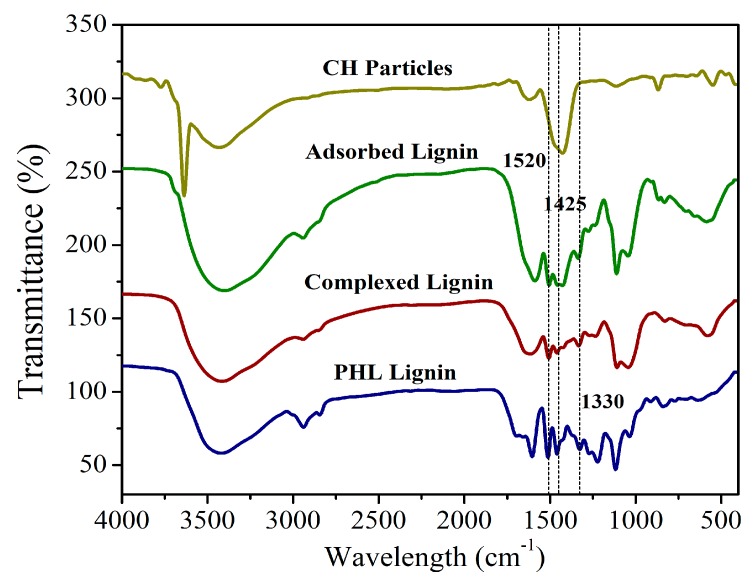
FTIR spectra of original CH particles, mixture of CH particles adsorbed and complexed by lignin, and original PHL lignin. Note: mixture of CH particles adsorbed lignin means the lignin adsorbed on CH particles in CH treatment process; mixture of complexed lignin means lignin precipitates obtained by CH solution treatment process; original PHL lignin means the lignin obtained by centrifugation of the original PHL.

**Figure 3 polymers-11-01558-f003:**
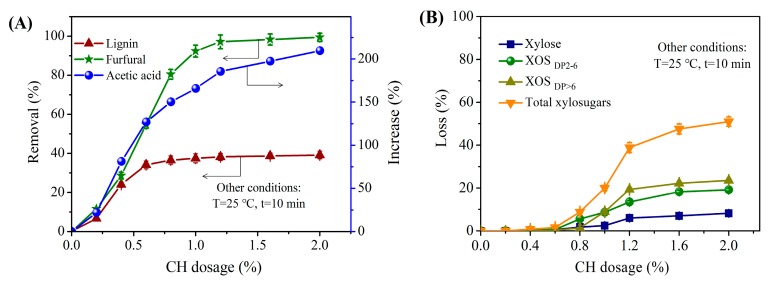
Influence of CH dosage on (**A**) the contaminant removal (furfural and lignin) and acetic acid increase of the original PHL and (**B**) the loss of xylose, XOS_DP2~6_, XOS_DP>6_, and total xylosugars.

**Figure 4 polymers-11-01558-f004:**
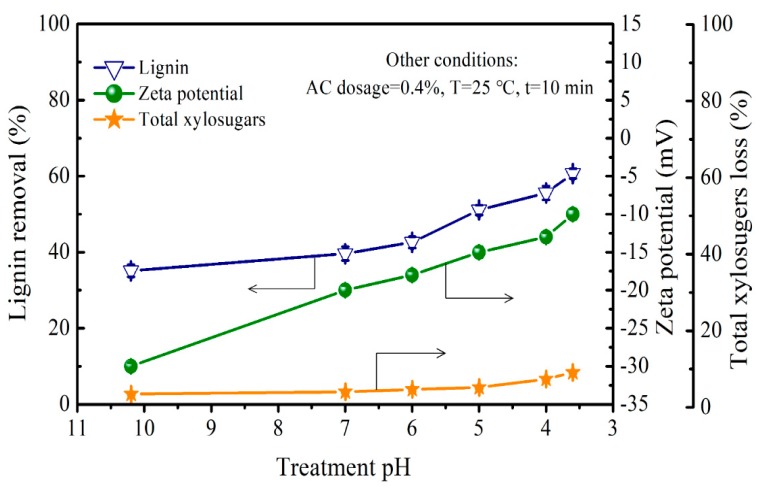
Influence of treatment pH on the lignin removal, total xylosugars loss, and zeta potential of AC.

**Figure 5 polymers-11-01558-f005:**
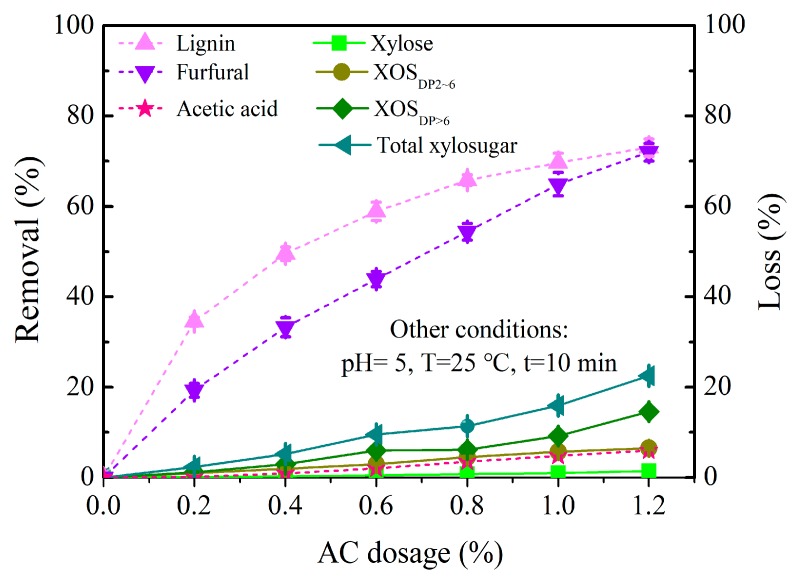
Influence of AC dosage on the removal of lignin, furfural, acetic acid and the loss of xylose, XOS_DP2~6_, XOS_DP>6_, and total xylosugars of the CH-treated PHL.

**Figure 6 polymers-11-01558-f006:**
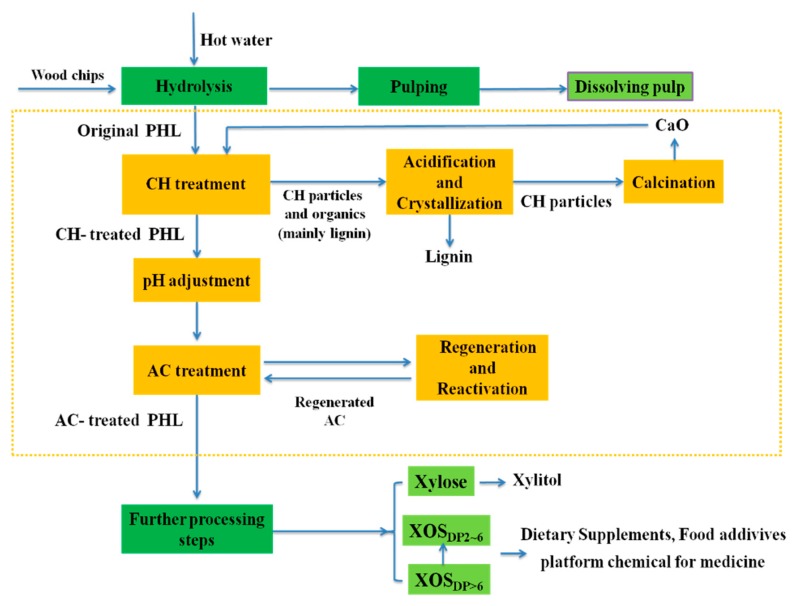
Proposed process for the purification and recovery of XOS from PHL.

**Table 1 polymers-11-01558-t001:** Chemical compositions of the original PHL (g/L).

Original PHL	Xylose	XOS_DP2~6_	XOS_DP>6_	Lignin	Acetic Acid	Furfural
Concentration	1.73	5.70	4.40	5.07	1.55	0.63

**Table 2 polymers-11-01558-t002:** Effect of CH existing forms and pH on lignin removal in CH treatment process.

CH Dosage (%)	pH	^a^ Lignin Removal Using CH in Solid and Liquid Forms (%)	^b^ Lignin Removal Using CH in Liquid Form (%)
0.2	6.6	6.7	3.1
0.4	9.0	24.0	5.0
0.6	10.0	34.3	9.2
0.8	11.0	37.1	9.8
1.2	11.5	38.0	10.1
2.0	12.0	38.6	10.2

Note: *a* means that CH powder was added into PHL to remove lignin by the formation of lignin-calcium absorption and lignin-calcium complexes. *b* means that the dissolved CH solution were added into PHL to remove lignin by the formation of lignin-calcium complexes at the same treating pH of CH treatment.

**Table 3 polymers-11-01558-t003:** Effect of combined treatment sequences of AC and CH on lignin and furfural removal and total xylosugars loss (%).

Treatment Process	Lignin Removal	Furfural Removal	Total Xylosugars Loss
AC (0.4%) + CH (0.4%)	48.3	46.2	5.4
AC (0.4%) + CH (0.6%)	49.2	69.3	6.0
CH (0.6%) + AC (0.4%)	66.9	70.1	5.9

**Table 4 polymers-11-01558-t004:** PHL compositions at different treatment stages (g/L).

PHL	Xylose	XOS	Total Xylosugars	Lignin	Furfural	Acetic Acid
Original PHL	1.73	10.10	11.83	5.07	0.63	1.55
CH treated PHL	1.69	10.05	11.74	3.33	0.28	3.50
CH+AC treated PHL	1.66	9.47	11.13	1.68	0.19	3.47
Removal (%)	4.0	6.2	5.9	66.9	70.1	-

Note: The optimal conditions were 0.6% dosage of CH (based on the weight of the original PHL) and 0.4% dosage of AC (based on the weight of the CH-treated PHL) in the combined treatment process.

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
