# Peer review of "Combined Treatments Consisting of Calcium Hydroxide and Activate Carbon for Purification of Xylo-Oligosaccharides of Pre-Hydrolysis Liquor"

_polymers, 2019, doi:10.3390/polym11101558_

Round 1
Reviewer 1 Report
Review (recommended major revision)
An article »Combined treatments consisting of calcium hydroxide and activate carbon for purification of xylo-oligosaccharides of pre-hydrolysis liquor« by Feng Xu et al. presents a research done in order to investigate the efficiency of the combined calcium hydroxide (CH) and activated carbon (AC) treatment towards the removal of lignin, furfural from pre-hydrolysis liquor (PHL) followed by optimal loss of targeted xylo-oligosaccharides (XOS). Authors report on successful removal of overall contaminants (lignin+furfural) with only moderate loss of total xylosugars which potentially could be applied for the purification and recovery of XOS from PHL.
The manuscript is poorly written, English should be strongly revised and improved. Some of the experimental results need a deeper scientific discussion, the experimental procedures have to be described more in detail. The results are valuable and contribute to the research area of residual biomass utilization, however the manuscript in the present form cannot be accepted. Due to the above mention comments I recommend the revision of this paper with possibility of re-submission after the corrections and replies of questions listed below.
Major issues that should be addressed.
Comment #1. English should be strongly revised and improved. There are quit few sentences especially in the Results and discussion.
Comment #2. “They can be transformed into fuel ethanol, levulinic acid, xylitol, etc. [3-5].” – The review of potential separation/downstream platforms is a bit poor, although this is the centre of manuscript. Some recent studies, for example Biomass & bioenergy, 2019, 120, 417, Chemical engineering journal, 2017, 330, 383, and Industrial & engineering chemistry research, 2019, 58(35), 16018 should be considered, expanding present modest review. The rest of manuscript’s introductory section seem to be ok.
Comment #3. Experimental: define the wood chip washing, identify solvent for preparation of CH solution, be more specific about ‘America’.
Comment #4. Results and discussion.
Figure 1 as well as figure caption should be improved as it is not really clear what authors want to emphasize. Figure should be informative enough to be able to stand for itself. Table 2: Units are missing for dosage. Column which is referred as CH treatment (%) data of lignin removal is listed. The title of column should be clear something as Lignin removal using CH in solid and liquid forms (%) etc. Furthermore, it is interesting solid and liquid CH treatment. Do authors mean the limited Ca(OH)2 dissolution in PHL? Some explanation is needed here. Figure 2: caption has to be corrected, specify.. FTIR spectra of samples. Row 242: The removal of lignin is mostly attributed.. do authors have any other process in mind beside the formation of lignin-calcium complexes and adsorption onto undissolved CH particles? Row 250: Add a brief explanation on beneficial effects of deacetylation towards enzymatic hydrolysis. Figure 3 A and B: scale should be from 0. Row 266: Higher Mw lignin has lower solubility in aqueous solution.. not really as solubility strongly depends on structure (terminal functional groups). Specify type of the lignin. Moreover, in reference 23 the adsorption process on activated carbon is discussed. Authors should clarify why the adsorption on CH surface is identical to the adsorption on activated carbon surface. Row 278: residual phenolic lignin? Is there any reason for such assignation as for instance significantly high number of aromatic OH groups? Row 296: Clarify which sample of CH-treated PHL was tested to determine the influence of activated carbon effect? Figure 5: A and B figures should be combined. Row 316: absorbed or adsorbed? Furthermore, polymer can not be adsorbed easily due to its lower solubility.. and in the reference 23 it is also explained: However, due to its high molecular weight (Table 1), the diffusion of lignin into the pores of AC was probably limited, which implies that AC had less accessible surface area for the adsorption of lignin than for the adsorption of furfural (Fig. 1). It is well known that the larger the polymers, the more insoluble they would be in aqueous solutions (Chinn and King, 1999; Sulaymonand Ahmed, 2008). Row 335: This paragraph is very confusing. After CH treatment lignin removal is 34.3% and additional lignin removal of 49.5% is achieved using activated carbon. Thus, the overall removal of lignin should make 83.8%. Authors should explain and clarify why the total lignin removal is considered to be 66.9%... while this number in abstract and in conclusions turns into 67.7%. Double-check and clarify. Row 351: More detailed explanation of every treatment in proposed process is needed.Comment #5. References: Regardless of the Comment #2, number of cited papers is too high for an original research article.
Author Response
Dear Reviewer:
Thank you so much for your constructive comments on our manuscript submitted to Polymers. Manuscript has been revised according to the comments. All changes have been highlighted in the revised manuscript, and point-by-point response to your comments have been uploaded. Here we submit the revised manuscript and response to reviewer comments. Thanks again!
Sincerely yours,
Dr. Guihua Yang

Reviewer 2 Report
After reading for review the article entitled "Combined treatments consisting of calcium 2 hydroxide and activate carbon for purification of 3 xylo-oligosaccharides of pre-hydrolysis liquor" for publication in the journal Polymers, I would like to highlight the following:
The meaning of the sentence (line 111) "CH powder existed in the undissolved CH particles and the dissolved CH solution during the treatment PHL process" It is mean that a part of the CH added in the PHL gets dissolved and another part not? It's clarified in results section, but not in the materials section that it where this concept appears for first time. The CH treatment was performed only for 10 minutes, why the authors choose this time? Some previous works estimate that it is a optimal time for this type of treatment? Some other experiments with different times were performed and 10 minutes was the optimal time? In the FTIR analysis, specify that the peak at 1520 cm-1 is associated to the vibration of C=C groups of the aromatic skeletal of lignin. In the Fig 3, the loss of xylose and the different XOS was analyzed. It was concluded that the loss of XOS was much more in CH treatment. It has any explanation? It is due to the greater charge interaction with CH particles? In line 259 authors said that XOSDP2-6 is higher that XOS>6 for increasing the CH content from 0.6 to 1%, however it was not observed in the graph, both samples produces an increase in a similar magnitude. It's more, the authors said that XOS>6 increase from 8.9 to 22.6 and XOSDP2-6 from 8.7 to 18.3, so the removal of DP2-6 is lower than >6. In line 268, the authors said that the optimum value is 0.6 due to the minimum xylosugars loss and maximum contaminants removal, please write the values of loss and removal for this CH dosage to clarify this affirmationFinally, to say that the article has been clearly written and explained and presents sufficient conclusions to justify the suitability of this treatment in comparison with others described in the bibliography (p-DADMAC and acidification process).
Author Response
Dear reviewer:
Thank you so much for your constructive comments on our manuscript submited to Polymers. Manuscript has been revised according to the comments. All changes have been highlighted in the revised manuscript, and point-by-point response to reviewer comments have been uploaded. hereby we submit the revised manuscript and response to reviewer comments. Thanks again!
Sincerely yours,
Dr.Guihua Yang

Round 2
Reviewer 1 Report
Review (accept)
Comments have been mostly addressed.